# Removal of Dye from a Leather Tanning Factory by Flat-Sheet Blend Ultrafiltration (UF) Membrane

**DOI:** 10.3390/membranes10030047

**Published:** 2020-03-18

**Authors:** Maryam Y. Ghadhban, Hasan Shaker Majdi, Khalid T. Rashid, Qusay F. Alsalhy, D. Shanthana Lakshmi, Issam K. Salih, Alberto Figoli

**Affiliations:** 1Membrane Technology Research Unit, Chemical Engineering Department, University of Technology, Alsinaa Street 52, Baghdad 10066, Iraq; 80209@uotechnology.edu.iq (M.Y.G.); 80007@uotechnology.edu.iq (K.T.R.); 2Department of Chemical Engineering and Petroleum Industries, AlMustaqbal University College, Babylon 51001, Iraq; hasanshker1@gmail.com (H.S.M.); Dr_IssamKamil@mustaqbal-college.edu.iq (I.K.S.); 3The Faculty of Pharmacy, Al-Kitab University, Altun Kupri Q565+Q8, Kirkuk, Iraq; lakshaluv@gmail.com; 4Institute on Membrane Technology, National Research Council (ITM-CNR), 87030 Rende (CS), Italy; a.figoli@itm.cnr.it

**Keywords:** polymer blend, membrane preparation, PPSU, PES, dyes, wastewater treatment

## Abstract

In this work, a flat-sheet blend membrane was fabricated by a traditional phase inversion method, using the polymer blends poly phenyl sulfone (PPSU) and polyether sulfone (PES) for the ultrafiltration (UF) application. It was hypothesized that adding PES to the PPSU polymer blend would improve the properties of the PPSU membrane. The effect of the PES concentration on the blend membrane properties was investigated extensively. The characteristics of PPSU-PES blend membranes were investigated using atomic force microscopy (AFM), scanning electron microscopy (SEM), contact angle measure, and contaminant (dye) elimination efficiency. This study showed that PES clearly affected the structural formation of the blended membranes. A considerable increase in the average roughness (about 93%) was observed with the addition of 4% PES, with a higher mean pore size accompanied by a rise in the pores’ density on the surface of the membrane. The addition of up to 4% PES had a significant influence on the hydrophilic character of the PPSU-PES membrane, by lowering the value of the contact angle (CA) (i.e., to 56.9°). The performance of the PPSU-PES composite membranes’ UF performance was systematically investigated, and the membrane pure water permeability (PWP) was enhanced by 25% with the addition of 4% PES. The best separation removal factor achieved in the current investigation for dye (Drupel Black NT) was 96.62% for a PPSU-PES (16:4 wt./wt.%) membrane with a 50% feed dye concentration.

## 1. Introduction

Ultrafiltration (UF) is an effective, low-pressure filtration technique for water treatment, especially to remove turbidity (e.g., suspended solids, bacteria, colloidal matter, and proteins). Particularly, the process has attracted a considerable amount of attention for water clarification and disinfection by size exclusion phenomena [1]. One of the most life-threatening environmental challenges is the removal of toxic contaminants and heavy metals from industrial effluents [2]. Due to the ever-increasing stringent regulations, lifetime expectancy issues, and economic benefits, industries are investing in effluent reuse/recycle techniques. Dyes play a vital part in coloring cellulosic membranes, such as cotton, and using reactive dyes causes dangerous environmental problems [3]. Textile industrial effluents are very difficult to treat because of the complexity of the dye molecules and their resistance to aerobic digestion, light, heat, and oxidizing agents. UF membrane processes can be a promising alternative for the removal of a variety of dye molecules from industrial effluents. Hence, a number of investigations have reported using UF and nanofiltration technologies in textile manufacturing industries [4].

Alventosa-deLara et al. [5] studied the influence of trans-membrane pressure (TMP) and cross-flow velocity (CFV) on the ultrafiltration (UF) ceramic membrane performance for various concentrations of dye (50–500 ppm). They concluded that the permeate flux increases with a higher TMP and CFV as well as with a low concentration of feed, up to a maximum flux of 0.267 m^3^/(m^2^ h). Moreover, the efficiency of the UF treatment was estimated by the dye rejection coefficient. Alventosa-deLara et al. [5] reported considerable dye rejection, regardless of the examined conditions, with steady-state rejection higher than 79.9% for 50 ppm and approximately 73.2% for 500 ppm. Moideen et al. [6] prepared a polysulfone-poly phenyl sulfone (PSF-PPSU blend UF membrane for Pb^2+^ and Cd^2+^ heavy metal removal, and the blend membrane displayed improved performance and hydrophilicity as well as an antifouling feature, as compared with the pristine PSF polymer membrane. Contaminant (e.g., heavy metals) rejection was completed by complexing the metal ions with polyaziridine, which has displayed a good rejection of 99.5% and 95.5% of Pb^2+^ and Cd^2+^, respectively. Since 2012, poly phenyl sulfone (PPSU) emerged as an excellent polymer material for membrane fabrication due to its advantageous characteristics, especially for NF application [7]. From the earlier reports about PPSU, membrane fabrication confirms that a key drawback was a lack of mechanical stability, which is crucial for separation efficiency and lifespan. To overcome this problem, in this study, we blended the polymer PPSU with polyether sulfone (PES) because of its beneficial properties: high performance, excellent mechanical intensity, a wide range of pore sizes, and good flexibility. The main purpose of this investigation was to enhance the characteristics of the PPSU membrane for dye elimination from simulated wastewater from the leather tanning industry.

In this work, PPSU membranes were prepared using a phase inversion method, with PES as a polymer blend material. The project investigated the effects of various concentrations of PES in the PPSU casting solution on the UF process performance. The structural morphology of the fabricated PPSU-PES membrane was analyzed via scanning electron microscopy (SEM), while the mean size of the pores and their distribution in the PPSU-PES membranes were examined with atomic force microscopy (AFM). Moreover, enhancement of membrane hydrophilicity was evaluated by measuring the contact angle. Finally, using the permeation flux and treatment of dye effluents, the efficiency of the PPSU-PES membranes was evaluated. 

In our previous work [8], a new approach to enhancing the dye removal performance of the membrane was suggested: using PES as a polymer blend with PPSU. Four concentrations of PES were investigated (from 0 to 6 wt.%), at a PPSU concentration of 20 wt.%. It was reported that the PES content aided in greatly increasing the hydrophilic character of the blend membrane (by about 15.2%), while also highly improving the blend membrane porosity (by approximately 280%). The blend membrane performance was improved four-fold more than when using the neat PPSU membrane. Therefore, in the present investigation, the blend of the two polymers (PPSU and PES) was evaluated as a membrane having the raw materials needed for dye rejection. The polymer blends were synthesized by changing the PPSU and PES concentrations from 20% to 16% and from 0% to 4% respectively (total polymers blend concentration was kept constant at 20%), via a phase inversion method, and studies were conducted to evaluate the impact of the blend composition on the membrane permeation flux and morphology, taking into consideration maintaining the rate of the dye rejection high.

## 2. Materials and Methods

### 2.1. Materials

PPSU (Radel R-5000 with 50 KDa average and 1.28 specific gravity) and PES (Radel) were purchased from Solvay Advanced Polymers (Brussels, Belgium). N-methyl-2-pyrrolidone (NMP) (C_5_H_9_NO) at 99.5% was used as a solvent, supplied by Sigma-Aldrich (St. Louis, MO, USA). Acid Black 210 dye (Drupel Black NT, Leeds, UK) was purchased through a local dealer in Iraq, with C_34_H_25_K2N_11_O_11_S_3_, λ max being the Acid Black molecular formula, and the structural formula is presented in Scheme 1.

### 2.2. Membrane Preparation

To remove moisture content, the polymer materials PPSU and PES were dried in an oven at 60 °C for 4 h. The dope solution was prepared by dissolving 20 wt.% dried PPSU in 80 wt.% NMP solvent. To obtain a homogeneous solution, PPSU was mixed in the NMP solvent using a magnetic mixer for 48 h, at 180 rpm and 45 °C. After the homogeneity of PPSU solution was obtained, the PES powder was added and mixed for a further 48 h. The influence of the PPSU:PES ratio shown in Table 1 on the properties of the membrane was studied. Utilizing a CX4 (MTV messtechnik oHG, Erftstadt, Germany) motorized film applicator under atmospheric conditions, the polymer solutions were cast at 200 µm thickness. The synthesized membranes were directly preserved in distilled water (the coagulation bath) at 25 °C for deposition. The product membranes were kept in distilled water for 2 to 3 days to eliminate the residual NMP. Finally, a glycerol solution (30 wt.%) was used to preserve the membranes for 48 h to keep the structure of the membrane from cracking and collapsing, and the membranes were dried at room temperature. In the test, three sheets of similar membranes were chosen to detect the features of the membrane and observe the UF process to obtain the mean value of the membranes’ performance for dye removal.

### 2.3. Membrane Surface Characteristics

#### 2.3.1. Contact Angle (CA) of the Membranes 

To determine the water contact angle (CA) of the PPSU-PES membranes, an optical CA device (110-O4W CAM, Tainan, Taiwan) was used. The mean value of the CA was specified in a progression of six evaluations for each of the various membrane surfaces.

#### 2.3.2. Fourier-Transform Infrared (FTIR) Spectroscopy

FTIR was employed for the quantitative and qualitative analysis of the organic groups, such as the molecular structure, molecular environment, and chemical bonding. Additionally, FTIR analysis was utilized to demonstrate the blending mode with a tensor device (Bruker, Optik GmbH, Ettlingen, Germany). Also, FTIR was utilized to investigate the interaction between two polymers and the miscibility of polymeric blends.

#### 2.3.3. Scanning Electron Microscopy (SEM)

Scanning electron microscopy (SEM) was used to observe the top surface and cross-sectional structural features of the flat-sheet membranes. SEM images were created with a TESCAN VEGA3 SB device (EO-Service, Kohoutovice / Czech Republic). Cross sections were readied for the SEM test by clipping the membranes in liquefied nitrogen. SEM pictures were recorded for the top surface as well as the cross-sectional area of the membrane using several magnification scales.

#### 2.3.4. Atomic Force Microscopy

PPSU-PES flat membranes were structurally characterized by displaying them using a large surface testing application with an atomic force microscope, model AA3000 (Angstrom Advanced Inc., Stoughton, MA, USA). The assessment included an estimation of the topography (the altitude and depth of the sample surface) and the side force (frictional force between the sample and tip, which resulted in a strain in the cantilever that can be seen in the photo detector’s left-right signal). The estimation of the deviation (the cantilever flexed towards the rise and fall of the sample topography) could be detected using the photo detectors’ up-down signal. For an upper and bottom surface of the entire flat-sheet membrane, the pore assessment dispersal was measured and a statistical pore size distribution for the entire surfaces was established by using IMAGER 4.31 programs.

#### 2.3.5. Porosity Measurement

Asymmetric microporous membranes were tested by evaluating the porosity and maximum pore size. Voids volume divided by the total porous membrane volume yields the membrane porosity, symbolized by ε. The overall membrane porosity was determined using the Equation below [9,10]:(1)$$ɛ=\frac{\rho_{m}}{\rho_{p}}$$

Membrane density was measured using the following Equation:(2)$$\;\rho_{m}\;=\frac{m}{l*w*\delta \;}$$
where $\rho_{m}$ is the density of the membrane (g/cm^3^),$\;\rho_{p}$ is polymer density (g/cm^3^), w is the width of the membrane (cm), m is the weight of the membrane sample (g), l is the membrane length (cm), and δ is the membrane thickness (cm). The density of the PPSU was 1400 kg/m^3^, while for PES, it was 1370 kg/m^3^.

#### 2.3.6. Mechanical Stability Measurement

A universal testing machine (UTM) (FH, Tinius Olsen) from the Department of Materials Research/Ministry of Science and Technology, Baghdad, was used to measure the tensile strength and percentage elongation of the PPSU-PES membrane. Using a suitable size (0.5 cm diameter × 1.5 cm length), the membranes were tested to determine their mechanical stability. 

#### 2.3.7. Membrane Performance

The performance of the PPSU-PES membranes as evidenced by the permeation flux and rejection were evaluated with cross-flow filtration tests utilizing membrane cells comprised of Teflon^®^. The following are the dimensions used in the study: effective membrane size (30 cm × 60 cm), total membrane module (80 cm × 50 cm × 45 cm), central bathtub with eight guides (0.1 cm in depth), and nozzle out with fast lock adapter (0.6 cm), which was sealed with steel screws. All tests were carried out at constant operating conditions: pressure of 1 bar and feed solution at room temperature in a cross-flow filtration system. The active membrane area was 1800 mm^2^, and the solution volume was 2000 cm^3^. Using the following Equation, pure water permeability (PWP) was estimated:(3)$$\mathrm{PWP}=\frac{V}{\;tAP}$$
where the membrane PWP is measured in L/(m^2^ h bar), *V* is the permeate volume (L), *P* is the trans-membrane pressure (bar), *t* is the time of the collected permeate (h), and *A* is the effective area of the membrane (m^2^). A solution of drupel black NT dye (Mw 938.017 g mol^−1^ and λ max = 460 nm), with a composition of 50, 75, and 100 ppm was used for the dye rejection measurement of each blend membranes. Rejection, R (%), of the contaminants was estimated using the following Equation [11]:(4)$$R\;(\%)=(1-\frac{Cp}{Cf})\times 100$$
where *C_f_* and *C_p_* are the drupel black NT dye concentrations (mg/L) of the feed and permeate, respectively. 

### 2.4. Solubility Parameter Difference (∆δ)

Solubility parameters of the 1-methyl-2-pyrrolidone (NMP) solvent, PPSU, and PES were obtained from the previous study [12]. The solubility factor variance between the PPSU and solvent as well as the solubility factor difference between the PES and solvent were estimated from the following Equations [13]:(5)$$\delta_{i,s}=\frac{X_{1}V_{1}\delta_{i,1}+X_{2}V_{2}\delta_{2.i}\;}{X_{1}V_{1}+X_{2}V_{2}},\;\;i=d,p,h$$
(6)$$\Delta \delta_{s-p}=\sqrt{\left[{\left(\delta_{d,s}-\delta_{d,p}\right)}^{2}+{\left(\delta_{s,p}-\delta_{p,p}\right)}^{2}+{\left(\delta_{h,s}-\delta_{h,p}\right)}^{2}\right]}$$
where the subscript *S* represents the NMP solvent, and the components are represented as follows: *p* is the polar, *d* is the dispersion, and h is the hydrogen-bonding.

## 3. Results and Discussion

### 3.1. Influence of PES on PPSU Membrane Morphology 

SEM pictures of the membrane’s top surface fabricated using PPSU with different concentrations of PES (i.e., 0%, 1%, 2%, 3%, and 4%) in the dope solution are illustrated in Figure 1. As can be seen from the figure, there is a considerable effect of the PES on the top surface of the PPSU-PES membranes. In Figure 1 (P1 and P2), the top surface was visualized as a dense layer with low pore density and small pore size, with 0% or 1% of PES in the dope solution. In contrast, with a PES concentration of 2%, the obtained top surface was less dense, as illustrated in Figure 1 (P3). Further increases in the PES concentration (e.g., 3% and 4%) caused large size pores on the upper surface and high pore density for the 4% in comparison with the 3% membrane. The addition of PES to the casting solution had a significant impact on the mechanism of the phase inversion operation that is followed when dealing with membrane morphology [1,14].

From Figure 1 (P1), it can also be seen that the cross-sectional structure of the membrane synthesized from pristine PPSU had wide sponge-like and finger-like structural layers. With 1% in the casting solution, the structure varied to become fingerlike with a long macro-void layer (Figure 1, P2). Thus, the addition of 1% PES resulted in obvious finger-like cavities in the sub-layer, with huge voids close to the top surface. In Figure 1, P3 shows that the addition of 2% PES to the polymer solution led to shorter macro-voids than with the addition of less PES and adding 2% PES also produced a decrease in the number of finger-like bundle arrangements. A further increase in the PES concentration of the polymer solution (i.e., up to 3%), caused a thin finger-void layer. The sponge-like structure was more substantive along the top surface layer, as illustrated in Figure 1 (P4). Further addition of PES to the casting solution (up to 4%) developed a membrane structure comparable to a membrane prepared from pristine PPSU (see Figure 1, P5). This behavior occurred because of the slowdown solvent (NMP)-nonsolvent (H_2_O) demixing operation between the casting solution and the H_2_O because of the effect of the PES polymer on the casting solution. 

Pakizeh et al. [15] pointed out that the macro-voids enlarge at low polymer concentrations in the dope solution and are restrained at higher concentrations of polymer. This phenomenon demonstrates that the presence of PES in a polymer casting solution significantly impacts the structure of the membrane. Moreover, with 3% PES in the casting solution, a finger-like layer was observed in the entire cross-section. This might result from the decline in the dispersion rate of the polymer particles because the PES has a higher surface energy. This causes particle agglomeration, which in turn leads to an increase in the NMP with the H_2_O interchange rate in the casting solution. Thus, it can be concluded that the mechanism of the membrane demixing process varied considerably by adding the PES to the polymer casting solution (i.e., blending it with PPSU polymer).

Table 2 presents the solubility factor variance between the membrane materials and the solvent, showing that the solubility factor variance between the PES and the NMP solvent was higher than that between the PPSU and the NMP solvent. This outcome suggests that the PES chains have less effect on the NMP than the PPSU on the NMP, which supports the need for a quick, diffusive flow of H_2_O and solvent (NMP).

### 3.2. Fourier-Transform Infrared (FTIR) Spectroscopy

Figure 2 displays the FTIR spectra of the PPSU-PES blend membranes synthesized with different blending percentages and pristine PPSU. Sulfone, sulfonamide, and quinone groups are present on the pristine PPSU membrane surface. Sulfonamide groups and sulfone form a negative charge on the membrane surface [16]. Figure 2 clearly shows that the pristine PPSU membrane peaks were comparable with the PPSU-PES membranes, except for the peak at 1157.28 cm^−1^, which was due to the presence of a C−O ester stretch. This peak was observed only on the membrane prepared from the pristine PPSU polymer. The peak disappeared when using PPSU-PES membranes because the chemical double bonds broke and connected the PPSU with the PES polymer. This in turn confirmed the good results of the mixing, which made the compound more stable. Other bonds (i.e., for C, H, and O) were repeatedly analyzed in a variety of membranes. Blending hydrophilic polymer into the dope solution can affect its phase inversion behavior, and thus, change the fabricated membrane properties [17,18].

### 3.3. Effect of PES on the PPSU Membrane Surface Roughness 

Figure 3 displays a three-dimensional AFM image of the upper surfaces of the PPSU membranes de-stained at various PES concentration in the PPSU dope solution. Figure 3 presents the various shapes of nodules in the top surface of the PPSU membranes synthesized with various PES concentrations. The nodular aggregates were integrated and composed a number of string-like structures as the concentration of PES rose to 3%. Further increases in the PES concentration yielded wide nodular aggregates dispersed on the surface of the membrane.

Table 3 displays the top surface of the PPSU flat-sheet membranes, which includes the mean roughness (Ra) (the mean value of the surface relative to the center plane), and the mean pore size, as estimated over a membrane area of 1 µm × 1 µm.

One of the most effective variables to enhance the antifouling capacity of membranes is the surface roughness. Thus, membranes with a smoother surface have better fouling impedance [19]. The PES polymer introduced to the membrane casting solution caused a notable increase in the surface roughness, from 23.6 nm for pristine PPSU, to 56.7 nm for 1% PES. The addition of more PES decreased the mean roughness to 13.59 nm (for 3% PES), fundamentally caused by the accumulation of extra PES on the surface of the membranes. Figure 3 supports these outcomes [20]. Based on the AFM assessments, the addition of 1% PES yielded a considerable enhancement in the mean roughness (by about 135%) compared to that synthesized from pristine PPSU. In contrast, the addition of 3% PES caused a significant decrease in the mean roughness (by 43%) compared with the mean surface roughness exhibited for pristine PPSU membranes. These outcomes refer to the tendency of PES to improve the antifouling membrane performance. A considerable enhancement of the mean roughness (by about 140%), with a comparable mean pore size and narrow pore size distribution, was gained with the addition of 1% of PES. Furthermore, it can be concluded that the presence of PES had a dominant influence compared with other influencing factors, such as the mean roughness, on the performance of the membrane. In addition, the outcomes were in line with those of the antifouling property noticed in other works [21,22]. Furthermore, from Table 3, it is clear that as the PES concentration in the casting PPSU solution rose to 1%, the mean pore size decreased, whereas further increases in the PES concentration caused the membrane to have a high mean pore size. These results were confirmed by the SEM images obtained of the membrane structure. Rajesh et al. [23] stated that an alteration in the mean roughness (Ra) of the membrane surface was related to an alteration in the pore size. Hong and He [24] concluded that the pore size diameter could be determined by the solvent and coagulation bath (water) exchange rate through the membrane formation.

Figure 4 shows the influence of the PES concentration on the pore size distribution of the PPSU and the top surfaces of the PPSU-PES membrane. The percentage of the accumulated distribution of the pore sizes was shifted to the right in comparison with the pristine PPSU membrane in Figure 4. There is a pronounced impact of the PES concentration on the pore size distribution, where raising the PES concentration in the PPSU polymer solution to 3% developed a wide range within the pore size distribution, from 0.025 to 1 µm, with a generally comparable accumulated distribution percentage of the pore size. The impact of the PES concentration on the pore size and the accumulated percentage of the pore size of the PPSU-PES dope solution have not been reported extensively in the literature. Furthermore, the pore size distribution narrowed with an increase in the PES content in the dope solution to 4%, as shown in Figure 5. The best pore size distribution was observed with 1% PES. The pore size distribution was narrow and frequently between 220 and 470 nm. These results are based on the membranes’ SEM analysis.

### 3.4. PPSU-PES Membranes Hydrophilicity

Figure 6 presents the contact angles (CA) of the PPSU membranes with diverse concentrations of PES (i.e., 0%, 1%, 2%, 3%, and 4%). The CA values decrease from 78.1° for pristine PPSU to 47.92° with a rise in PES concentrations in the casting solution to 1%. On the contrary, further increases in the PES concentration raised the CA value to 56.9° for 4% PES concentrations. Therefore, we conclude that PES (up to 4%) as a polymer blend has a positive influence on the hydrophilicity of the PPSU-PES membrane surface, with a reduction in the CA value by about 25%.

The PPSU-PES blend composite membranes appeared to have a lower contact angle (P5, 56.9°) than that of the pristine PPSU membrane (78.1°). The contact angle estimation was correlated to the surface hydrophilicity, with a lower contact angle indicating the effect of adding PES to the PPSU casting blend composition. This behavior is due to the hydrophilic feature of the PES polymer. 

The addition of the PES as a polymer blend to PPSU was due to the tendency of PES to form a porous structure at the membrane surface and thus improves the value of the hydrophilic character of the surface because surface porous features were one of the profound factors that affects water contact angle. The porous structure that appeared at the membrane blend surfaces presented in Figure 1 confirm this phenomenon.

### 3.5. Effects of PES on Membrane Thickness and Porosity

Membrane thickness was evaluated using the SEM technique, as shown in Figure 7. It can be seen that raising the PES content in the polymer solution caused an increase in membrane thickness. Increasing the concentration of the PES in the dope solution caused an increase in the solution viscosity, which led to a slowdown of the demixing process between the coagulation bath (water) and the solvent (NMP) through the membrane formation, which produced a maximally thick membrane with a sponge-like structure.

Increasing the PES to 3% or 4% caused a trend that slightly decreased the membrane thickness. This may be due to the hydrophilic nature of the PES in the PPSU matrix, which enhanced the solution’s thermodynamic instability in the phase inversion process and also increased porosity. When the PES concentration increased from 1% to 2%, there was a sudden jump in the porosity value, possibly because of the increase in the number of finger-like shapes. Finger-like and macro-voids shapes in the blend membrane caused a rise in porosity, which increased with the addition of PES as the finger-like structure of the membranes improved the pore interconnectivity. The results prove that PES molecules affected the distribution of PPSU in the membrane matrix. Further increases in the PES concentration had little influence due to agglomerations at higher loading, which resulted in a slowdown of the demixing rate of the solvent and nonsolvent (water) through the phase inversion operation, which caused a low porosity as well as minimized the mean pore radius in the membrane. Also, the difference in interaction parameters of two polymers with solvent may lead to aggregation of PES in PPSU-PES at high PES loading during membrane formation, and the results presented in Table 2 confirm this behavior. Thus, it can be realized that the relation between the membrane thickness and porosity were influenced by the polymer properties and its sympathy and interaction parameters between the solvent and polymer. Besides, increasing the amount of the hydrophilic polymer could affect the trends in membrane thickness and porosity.

### 3.6. Performance of the PPSU-PES Membrane

Figure 8 shows the influence of diverse PES concentrations in the PPSU dope solution on the pure water permeability (PWP) of PPSU-PES membranes. The PWP was 23 (L/m^2^. h) for the pristine PPSU membrane, whereas adding PES to the dope solution caused the PWP to improve to 78.4, 62.9, and 60.69 (L/m^2^. h) for the PPSU-PES membranes with 1%, 2%, and 3% PES in the polymer solution, respectively. On the other hand, increasing the PES concentration beyond 4% led to an increase in the PWP, to about 67.25 (L/m^2^. h). This property influences the contact angle, pore size, and pore size distribution, which significantly impacts the PWP. AFM analysis outcomes and CA values previously displayed in this paper support this conclusion. However, the hydrophilicity seems to control more than the pore size, which is not congruent with the outcomes of Li et al. [25]. Furthermore, some researchers consider that hydrophilicity plays a key role in the PWP of membranes [26,27]. Moreover, it can be realized from Figure 8 that the flux values for P3, P4, and P5 were convergent within experimental errors. However, the approximately similar porosity of these three membranes as clearly depicted in Figure 6 seems to control the results of the permeation flux.

### 3.7. Effect of PES on Mechanical Properties

Although PPSU membranes exhibit thermal stability and comprehensive strength properties, they consist of a tough polymer with a high glass-transition temperature. Moreover, PPSU membranes are not suitable in terms of their mechanical properties, as they have a bulky structure, and the binding force of their inside membranes structure is low. This study measured the ultimate tensile strength and elongation at break of the prepared membranes. Results are shown in Figure 9, where, in general, the prepared membranes display enhanced mechanical features after PES blending. This improvement resulted from the PES having diffused uniformly within the PPSU polymer during the preparation of the casting solution, especially for 1% and 2% PES loading, which in turn improved the interconnectedness within the blended membrane matrix. In Figure 9, where the prepared membrane with 2% PES (P3) displayed maximum tensile strength due to the higher membrane thickness, approximately lower porosity and shorter macro-voids also produced a decrease in the number of finger-like bundle arrangements debated in parts entitled ‘‘Influence of PES on PPSU membrane morphology’’. When the PES layer occupied the void space in the membranes, the PES layer glued the membrane at points of overlap. The mechanical features might be further enhanced by building an additional PES layer onto the PPSU matrix.

The tensile strength and elongation of the prepared membranes rely on the morphological structure, thickness, and porosity of the membrane, which were affected by PES content in casting solution, as shown in Figure 1, and debated in parts entitled ‘‘Influence of PES on PPSU membrane morphology’’ and ‘‘Effects of PES on membrane thickness and porosity”. In general, macro-voids in wall of the membranes were unfavorable because they display lacking intensity regions (weak points). Therefore, under high operating pressures, this type of membrane may fail. Consequently, controlling the structure of the membrane to develop its performance, a settlement between the strength property and permeability has to be obtained.

### 3.8. Dye removal Efficiency of the PPSU-PES Membranes

The feasibility of using the PPSU-PES membranes was estimated by conducting the UF experiments with different concentrations (i.e., 50, 75, and 100 ppm) of dye solutions as feeds. The concentrations of dye in the feed and permeate streams were measured using a UV-vis spectrophotometer (Pharo 300, Merck, Darmstadt, Germany). The dye rejection R (%) was evaluated using Equation (4). Figure 10 shows the rejection of three dye concentrations as feed solutions and the influence of PES polymer in the polymer solution on the dye (Drupel Black NT) rejection by the PPSU membranes. From Figure 10, it is obvious that the dye rejection increased with an increasing concentration of PES in the dope solution. Also, no significant difference in dye rejection values was obtained for membranes prepared from P2 to P4 within experimental errors in Figure 10. However, the slight difference in mean pore size and thickness of these three membranes (see Table 3 and Figure 7) was the real reason behind these results. This occurred because the addition of PES enhanced the membrane morphology and structure, including the pore size, pore size distribution, contact angle, and surface roughness [7].

The permeation fluxes were greatly enhanced with high dye removal efficiency of the simulated wastewater by the addition of PES as a polymer blend. Figure 11 shows the influence of the PPSU:PES ratio on the morphological structure of the membrane, and therefore, on the permeation flux and separation performance of the membrane.

## 4. Conclusions

PPSU-PES blend membranes of different compositions were fabricated and their rejection behavior of C.I. (AB210) dye (Drupel Black NT) was analyzed. The effect of the PES concentration on the membrane morphology was evaluated using SEM and AFM techniques. Synthesized membranes exhibited an asymmetric structure with an upper dense layer and a bottom porous layer, with finger-like projections that increased with the addition of PES polymer to the PPSU casting solution. A considerable increase in the average roughness (of about 93%) was observed with the addition of 4% PES, with a higher mean pore size accompanied by a rise in the surface pore density of the membrane. The addition of 4% PES also improved the hydrophilicity of the blend membrane. Membranes with 0% PES content appeared to exhibit a lower dye rejection than membranes with a PES additive because the mean pore size and porosity of the membranes increased with the addition of PES. The mechanical properties of the blended membranes were highly improved by the addition of PES, especially those with 2% PES content. PPSU-PES blending polymers have dramatically enhanced the membrane performance, including the permeation flux and dye rejection. According to the reassuring results found, especially the developed dye rejection performance demonstrated by 4% PES (P5), future work might be addressed to investigate the effect of PES loading higher than 4% on separation performance of the membrane.

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
