# Peer review of "Removal of Dye from a Leather Tanning Factory by Flat-Sheet Blend Ultrafiltration (UF) Membrane"

_membranes, 2020, doi:10.3390/membranes10030047_

Round 1

Reviewer 1 Report

  1. The last two paragraphs under Introduction must be reorganized, since the present manuscript has a high percentage of similarity to a cited (Al-Ani et al., 2019) publication of the corresponding author, in order to state what is need of and new finding from the present study compared to the former work.
  2. Section 2.2 is incomplete because it does not mention how the cast liquid layer was dried and how PES was incorporated in the PPSU matrix.

  3. Section 2.3.7 must provide structural / composition information of drupel black NT dye.
  4. For the binary polysulfone blend in question, the thermal analysis by differential scanning calorimetry or dynamic mechanical testing of the pure PPSU and the most effective PPSU-PES , e.g. P5, is critical to understand how the chain-segment motions of PPSU are affected by the presence of PES in proximity vicinity. This interaction affects membrane porous traits. On contrary, FT-IR and water-contact angle are not very necessary.

  5. Table 2, the mean pore sizes represent only both surfaces but not what are present in the bulk. The large pore channels shown in the right column of Fig. 1 are apparently not suitable to the microfiltration.

  6. Figure 2 is actually superfluous. Figure 6 means little value because surface porous features affects water contact angle more profoundly than the PES content.

  7. Figure 4 and 5: How was vol % vs pore size determined? This type of profile could be obtained from Hg porosimetry or BET adsorption analysis.

  8. Figure 9 has to be expressed using permeance rather than flux so that it could be better correlated with dye rejection performance.

  9. Figure 10, what is the intrinsic relation of this test with the separation capability of membrane?

  10. Figure 11 shows that only membrane P5 demonstrates improved dye rejection performance and the rest PPSU-PES membranes are not distinguishable from P1. This implies a need to check a membrane with a higher PES loading than P5.   

Author Response

Dear Editor

I would like to submit a revised form manuscript entitled “Removal of dye from a leather tanning factory by flat-sheet blend ultrafiltration (UF) membrane" for consideration for publication in the Membranes. This article is revised according to the reviewer comments. I appreciate the effort of the reviewer to improve our article, thank you. Please answer to the reviewer comments presented below.

Your consideration for this manuscript with revised form is highly appreciated.

Sincerely

Prof. Dr. Qusay F. Alsalhy

Director of Membrane Technology Research Unit

Chemical Engineering Department

University of Technology,

Alsinaa Street No. 52

Baghdad, Iraq

Email:qusay_alsalhy@yahoo.com

80006@uotechnology.edu.iq

Answer to the reviewer's comments:

Reviewer 1

  1. The last two paragraphs under Introduction must be reorganized, since the present manuscript has a high percentage of similarity to a cited (Al-Ani et al., 2019) publication of the corresponding author, in order to state what is need of and new finding from the present study compared to the former work.

Answer: We agree with the reviewer comment, we see that there is a lack in the objectives paragraph to express the new finding of the present work.

The difference between the present work and the former one was with the concentration of PPSU, in previous work the concentration was kept constant at 20% with increasing of PES up to 6% (the total polymers concentration was 26%), while in the present work the PPSU concentration was decrease from 20 to 16% with increasing of PES up to 4% (the total polymers concentration was 20%). The new finding from the present study is to increasing the permeation flux with decreasing of polymers concentration in casting solution taking into consideration maintaining the high rate of dye rejection.

Therefore, we revised the following paragraph:

"The polymer blends were synthesized by changing the PPSU and PES concentrations from 20 to 16% and from 0 to 4%, respectively (total polymers blend concentration was kept constant at 20%) via a phase inversion method, and studies were conducted to evaluate the impact of the blend composition on the membrane permeation flux and morphology taking into consideration maintaining the rate of the dye rejection high."

  1. Section 2.2 is incomplete because it does not mention how the cast liquid layer was dried and how PES was incorporated in the PPSU matrix.

Answer: We agree with the reviewer comment, therefore we added the following highlighted red sentences:

"PPSU was mixed in the NMP solvent using a magnetic mixer for 48 h, at 180 rpm and 45°C. After the homogeneity of PPSU solution was obtained the PES powder was added and mixed for farther 48 h. The influence of the PPSU: PES ratio on the properties of the membrane was studied. Utilizing a CX4 MTV messtechnik (Germany) motorized film applicator under atmospheric conditions, the polymer solutions were cast at 200 µm thickness. The synthesized membranes were directly preserved in distilled water (the coagulation bath) at 25°C for deposition. The product membranes were kept in distilled water for 2-3 days to eliminate the residual NMP. Finally, a glycerol solution (30 wt. %) was used to preserve the membranes for 48 h to keep the structure of the membrane from cracking and collapsing. Finally, the membranes were dried at room temperature."

  1. Section 2.3.7 must provide structural/composition information of drupel black NT dye.

Answer: Thanks for this comment; we added the following sentence in section 2.3.7 and the structural formula was mentioned and presented in scheme 1in materials section:

"A solution of drupel black NT dye (Mw 938.017 g mol-1 and λ max =460 nm), with 50, 75 and 100 ppm was used for the dye rejection measurement of each blend membranes."

  1. For the binary polysulfone blend in question, the thermal analysis by differential scanning calorimetry or dynamic mechanical testing of the pure PPSU and the most effective PPSU-PES, e.g. P5, is critical to understand how the chain-segment motions of PPSU are affected by the presence of PES in proximity vicinity. This interaction affects membrane porous traits. On contrary, FT-IR and water-contact angle are not very necessary.

Answer: Actually we are agreeing with the reviewer comment that the differential scanning calorimetry (DSC) testing of pure PPSU and the most effective PPSU-PES, to understand how the chain-segment motions of PPSU are affected by the presence of PES in proximity vicinity. However, we would like to mention that one of the main methods to select the polymer blends is the solubility parameter difference. The blends have excellent miscibility when the solubility parameter difference of the blends with the solvent is approximately similar at various polymer-polymer blends ratio. Moreover, it is evident that the glass transition temperature (Tg °C) of a polymer measure by differential scanning calorimetry (DSC) is an important criterion for the miscibility of polymers. For an immiscible polymer blend, every polymer should possess its own (Tg). Whereas, for a fully miscible polymer blend, a single glass transition temperature (Tg) should be obtained. Fourier Transform Infrared (FTIR) spectroscopy was another important criterion utilized to study the interaction between two polymers and the miscibility of polymeric blends. From FTIR analysis the miscibility of polymeric blends is confirmed because a few shifts are seen in FTIR spectra. In the current study it was used the solubility parameter difference and FTIR criterion to measure the miscibility of the polymers blend.

According with what mention above, we would like to mention here that, as the reviewer know that most universities announced that their official working hours had been suspended because of the Corona virus, so it became difficult to measure the glass transition temperature (Tg °C) of a polymers by differential scanning calorimetry (DSC) and to understand how the chain-segment motions of PPSU are affected by the presence of PES.

However, we added the following sentences to section 2.3.22.3.2. Fourier-transform infrared (FTIR) spectroscopy:

"Also, FTIR was utilized to investigate the interaction between two polymers and the miscibility of polymeric blends."

  1. Table 2, the mean pore sizes represent only both surfaces but not what are present in the bulk. The large pore channels shown in the right column of Fig. 1 are apparently not suitable to the microfiltration.

Answer:

It is obvious that the resulting membrane consists of a thin porous or dense skin about 0.1–1.0 µm thick, called the permselective layer, formed over a much thicker microporous layer that provides support for the skin. Therefore, the measurement of pore size at the top and bottom surfaces of such permselective layer was carried out and it was suitable for ultrafiltration membrane. The thicker microporous layer shown in the right column of Figure 1 was only provides support for the skin or permselective layer.

  1. Figure 2 is actually superfluous. Figure 6 means little value because surface porous features affects water contact angle more profoundly than the PES content.

Answer: We agree with the reviewer comment, therefore we added the following sentences page 11 line 320:

"The addition of the PES as a polymer blend to PPSU was due to the tendency of PES to form a porous structure at the membrane surface and thus improves the value of the hydrophilic character of the surface because surface porous features were one of profoundly factors that affects water contact angle and the porous structure appeared at the membrane blend surfaces presented in Figure 1 confirm this phenomenon."

  1. Figure 4 and 5: How was vol % vs pore size determined? This type of profile could be obtained from Hg porosimetry or BET adsorption analysis.

Answer: We agree with the reviewers comment; we added in page 4, line 133, the following sentence to describe the determine the Vol.% vs pore size:

"For an upper and bottom surface of the entire flat-sheet membrane, the pore assessment dispersal was measured and a statistical pore size distribution for the entire surfaces was established by using IMAGER 4.31 programs."

  1. Figure 9 has to be expressed using permeance rather than flux so that it could be better correlated with dye rejection performance.

Answer: Figure 9 was not the results of the current work, for this reason we did not present its discussed in the manuscript. Therefore, we delete it from the paper. We are apologize to the reviewers.

  1. Figure 10, what is the intrinsic relation of this test with the separation capability of membrane?

Answer: It is very important comment; therefore, we added the following paragraph in page 14, line 379:

"The tensile strength and elongation of the prepared membranes rely on the morphological structure, thickness and porosity of the membrane, which were affected by PES content in casting solution, as shown in Figure 1, and debated in parts entitled ‘‘Influence of PES on PPSU membrane morphology’’ and ‘‘Effects of PES on membrane thickness and porosity. In general, macrovoids in wall of the membranes were unfavorable because they display lacking intensity regions (weak points). Therefore, under high operating pressures this type of membrane may failure. Consequently, controlling structure of the membrane to develop its performance, a settlement between the strength property and permeability has to be obtained."

  1. Figure 11 shows that only membrane P5 demonstrates improved dye rejection performance and the rest PPSU-PES membranes are not distinguishable from P1. This implies a need to check a membrane with a higher PES loading than P5.

Answer: It is important comment extracted by the reviewer from the results of the dye rejection performance. We clearly observed from Figure 11 that P4 also demonstrates improved dye rejection performance. Therefore, we added the following paragraph in the end of the conclusions:

 "According to the reassuring results found, especially the developed dye rejection performance demonstrates by 4% PES (P5), future work might be addressed to investigate the effect of higher PES loading than 4% on separation performance of the membrane."

Important note: Comments from 11 to 20 are repeated comments (similar to comments 1-10)

Reviewer 2 Report

The manuscript by Ghadhban et al. shows a novel method to blend polyphenyl sulfone (PPSU) with Polyether sulfone (PES) to prepare ultrafiltration (UF) membranes for the removal of a dye from leather tanning industry. The manuscript provides a thorough characterization of the blended polymer membranes and provides a good discussion on the comparison of properties of unmodified and modified membranes. However, there are certain areas of the manuscript where it can be improved. I recommend the manuscript should be reconsidered for publication after a major revision. My comments are as follows:

  • There are several grammatical errors in the manuscript. I recommend proofreading again before resubmitting the manuscript.
  • The work is similar to the work done by Al-Ani et al. (2019). Although the authors have mentioned that this manuscript evaluated the blend of the polymers as a membrane having the raw materials needed for dye rejection, this doesn’t provide a clear idea about the novelty of the process and the difference between the previous work and this work. Please provide a detailed and thorough discussion to give the reader a clear idea.
  • Page 4, line no. 133: Please mention what ρp
  • Page 7, line no. 206: The discussion of table 2 in the manuscript and contents of table 2 don’t match.
  • Section 3.2, line no. 217: It would be helpful to point out the peak at 1157.28 in figure 2.
  • Page no. 9, line no. 238: Table 2 should be added after the discussion in lines 238-240. Also, please add Rms values in the table.
  • Line no. 246: Please correct “figure 2” to “figure 3”.
  • Section 3.4: Although the contact angle decreased from P1 to P2, why was there an increase in contact angle as the concentration of PES increased in the membrane?
  • Page no. 13, line no. 314: Is there a way to confirm the agglomeration of PES using the characterizations done in the study? Does FTIR show more peaks corresponding to the functional groups in PES as the aggregation increased?
  • Figure 8: The flux values are approximately similar for P3, P4, and P5 within experimental errors and they are lower than P1. This shows that increasing the PES concentration more than 1% doesn’t have a significant effect on PWP. Please add a discussion on this.
  • Please add a discussion for figure 9 in the manuscript.
  • Page 14, line no. 339: “The binding force of their inside membranes is low”. Please elaborate what “inside membranes” means.
  • Page no 14, line no. 342: It is mentioned before (Page no. 13, line 314) that there are aggregations of PES. However, line no. 342 mentions that PES is diffused uniformly within the PPSU polymer. This makes the explanation somewhat confusing. Please elaborate more on this.
  • Figure 10: Is there a reason the tensile strength is maximum for P3? Please change the Y-axis label from ‘Tensile’ to ‘Tensile strength’.
  • Page 15, Table 3: Please move the table to AFM section.
  • Page 15, line no. 356. Please change ‘figure 10’ to figure 11. Please make sure all the diagrams and tables are numbered properly in the captions as well as in the discussions.
  • Figure 11: There is no significant difference in the rejections obtained from P2 to P4. The rejections are approximately similar within experimental errors. Please elaborate more on this.
  • Section 4.: It would be helpful to mention or propose an optimized percentage for PES to blend with PPSU.
  • Reference 1: Please correct the name of the article. Is it nanofiltration or ultrafiltration?

Author Response

Dear Editor

I would like to submit a revised form manuscript entitled “Removal of dye from a leather tanning factory by flat-sheet blend ultrafiltration (UF) membrane" for consideration for publication in the Membranes. This article is revised according to the reviewer comments. I appreciate the effort of the reviewer to improve our article, thank you. Please answer to the reviewer comments presented below.

Your consideration for this manuscript with revised form is highly appreciated.

Sincerely

Prof. Dr. Qusay F. Alsalhy

Director of Membrane Technology Research Unit

Chemical Engineering Department

University of Technology,

Alsinaa Street No. 52

Baghdad, Iraq

Email:qusay_alsalhy@yahoo.com

80006@uotechnology.edu.iq

Answer to the reviewer's comments:

Reviewer 2

The manuscript by Ghadhban et al. shows a novel method to blend polyphenyl sulfone (PPSU) with Polyether sulfone (PES) to prepare ultrafiltration (UF) membranes for the removal of a dye from leather tanning industry. The manuscript provides a thorough characterization of the blended polymer membranes and provides a good discussion on the comparison of properties of unmodified and modified membranes. However, there are certain areas of the manuscript where it can be improved. I recommend the manuscript should be reconsidered for publication after a major revision. My comments are as follows:

  • There are several grammatical errors in the manuscript. I recommend proofreading again before resubmitting the manuscript.

Answer: The manuscript was proofreading again please see the attached English editing file.

  • The work is similar to the work done by Al-Ani et al. (2019). Although the authors have mentioned that this manuscript evaluated the blend of the polymers as a membrane having the raw materials needed for dye rejection, this doesn’t provide a clear idea about the novelty of the process and the difference between the previous work and this work. Please provide a detailed and thorough discussion to give the reader a clear idea.

Answer: The difference between the present work and the former one was with the concentration of PPSU, in previous work the concentration was kept constant at 20% with increasing of PES up to 6% (the total polymers concentration was 26%), while in the present work the PPSU concentration was decrease from 20 to 16% with increasing of PES up to 4% (the total polymers concentration was 20%). The new finding from the present study is to increase the permeation flux with decreasing of polymers concentration in casting solution taking into consideration maintaining the high rate of dye rejection.

Therefore, we revised the following sentences:

"The polymer blends were synthesized by changing the PPSU and PES concentrations from 20 to 16% and from 0 to 4%, respectively (total polymers blend concentration was kept constant at 20%) via a phase inversion method, and studies were conducted to evaluate the impact of the blend composition on the membrane permeation flux and morphology taking into consideration maintaining the rate of the dye rejection high."

  • Page 4, line no. 133: Please mention what ρp

Answer: we added the definition of ρp

  • Page 7, line no. 206: The discussion of table 2 in the manuscript and contents of table 2 don’t match.

Answer: We delete the root mean square of Z values (Rms), from section 3.3., Effect of PES on the PPSU membrane surface roughness, because it is not presented in Table 2.

  • Section 3.2, line no. 217: It would be helpful to point out the peak at 1157.28 in figure 2.

Answer: Thanks to the reviewer; we pointed out the peak at 1157.28 cm-1 in Figure 2 please see the updated Figure 2.

  • Page no. 9, line no. 238: Table 2 should be added after the discussion in lines 238-240. Also, please add Rms values in the table.

Answer: we moved Table 2 after discussion and we added Rms values in Table 2.

  • Line no. 246: Please correct “figure 2” to “figure 3”.

Answer: We revised Figure 2 to Figure 3.

  • Section 3.4: Although the contact angle decreased from P1 to P2, why was there an increase in contact angle as the concentration of PES increased in the membrane?

Answer: We agree with the reviewer comment, therefore we added the following sentences in page 11 line 320:

"The addition of the PES as a polymer blend to PPSU was due to the tendency of PES to form a porous structure at the membrane surface and thus improves the value of the hydrophilic character of the surface because surface porous features were one of profoundly factors that affects water contact angle and the porous structure appeared at the membrane blend surfaces presented in Figure 1 confirm this phenomenon."

  • Page no. 13, line no. 314: Is there a way to confirm the agglomeration of PES using the characterizations done in the study? Does FTIR show more peaks corresponding to the functional groups in PES as the aggregation increased?

Answer: It is excellent comment:

 SEM and FTIR for example can be used to confirm the agglomeration of any polymer to another in polymer blend if the two polymers are immiscible or physical compatible obtained between them as reported by previous work shown in reference below.

"Qusay F. Alsalhy, Hollow fiber ultrafiltration membranes prepared from blends of poly (vinyl chloride) and polystyrene, Desalination 294 (2012) 44–52."

In the current study miscible blend was obtained according to FTIR analysis and SEM images. For example, from FTIR analysis the miscibility of polymeric blends is confirmed because a few shifts are seen in FTIR spectra, and in SEM images, there are no any spots or points of PES at the surface of the PPSU-PES membrane. The difference in interaction parameters of two polymers with solvent may lead to aggregation of PES in PPSU-PES at high PES loading.

Therefore we added the following sentence to page 13 line 314 to confirm the increasing of the aggregation of PES in PPSU-PES membranes.      

"Also, the difference in interaction parameters of two polymers with solvent may lead to aggregation of PES in PPSU-PES at high PES loading during membrane formation and the results presented in Table 2 confirm this behavior."

  • Figure 8: The flux values are approximately similar for P3, P4, and P5 within experimental errors and they are lower than P1. This shows that increasing the PES concentration more than 1% doesn’t have a significant effect on PWP. Please add a discussion on this.

Answer: We added the following discussion:

"Moreover, it can be realized from Figure 8 that the flux values for P3, P4, and P5 were convergent within experimental errors. However, the approximately similar porosity of these three membranes as clearly depicted in Figure 6 seems to control the results of the permeation flux."

  • Please add a discussion for figure 9 in the manuscript.

Answer: Figure 9 was not the results of the current work, for this reason we did not present its discussed in the manuscript. Therefore, we delete it from the paper. We are apologize to the reviewers.

  • Page 14, line no. 339: “The binding force of their inside membranes is low”. Please elaborate what “inside membranes” means.

Answer:  We revised the sentence as follows to be clear for the reader:

"the binding force of their inside membranes is low" Revised to "the binding force of their inside membranes structure is low."

  • Page no 14, line no. 342: It is mentioned before (Page no. 13, line 314) that there are aggregations of PES. However, line no. 342 mentions that PES is diffused uniformly within the PPSU polymer. This makes the explanation somewhat confusing. Please elaborate more on this.

Answer: In page no. 13, line 314 it was mentioned that "Further increases in the PES concentration had little influence due to agglomerations at higher loading of PES" thus, for 4 and 5% PES loading the membrane mechanical properties were reduced. Therefore, in order to make the discussion of this behavior clear for the reader we revised the following sentence:

"….diffused uniformly within the PPSU polymer during the preparation of the casting solution especially for 1 and 2% PES loading,…."

  • Figure 10: Is there a reason the tensile strength is maximum for P3? Please change the Y-axis label from ‘Tensile’ to ‘Tensile strength’.

Answer: We added the following reason in section 3.7 Effect of PES on mechanical properties:

"In Figure 9, where the prepared membrane with 2% PES (P3) display maximum tensile strength due to the higher membrane thickness, approximately lower porosity and shorter macro-voids also produced a decrease in the number of fingerlike bundle arrangements debated in parts entitled ‘‘Influence of PES on PPSU membrane morphology’’

  • Page 15, Table 3: Please move the table to AFM section.

Answer: We revised Table 3 to Table 2 and moved to Page 7 line no. 219 SEM section because it was the first referred there.

  • Page 15, line no. 356. Please change ‘figure 10’ to figure 11. Please make sure all the diagrams and tables are numbered properly in the captions as well as in the discussions.

Answer: We revised number and location of all Tables and Figures in the text and caption carefully, thanks for this comment

  • Figure 11: There is no significant difference in the rejections obtained from P2 to P4. The rejections are approximately similar within experimental errors. Please elaborate more on this.

Answer: We agree with the reviewer comment, therefore we added the following paragraph:

"Also, no significant difference in dye rejection values was obtained for membranes prepared from P2 to P4 within experimental errors in Figure 10. However, the slight difference in mean pore size and thickness of these three membranes (see Table 3 and Figure 7) was the real reason behind these results."   

  • Section 4.: It would be helpful to mention or propose an optimized percentage for PES to blend with PPSU.

Answer: Thanks for this nice comment; therefore we added the following sentence:

According to the reassuring results found, especially the developed dye rejection performance demonstrates by 4% PES (P5), future work might be addressed to investigate the effect of higher PES loading than 4% on separation performance of the membrane.

  • Reference 1: Please correct the name of the article. Is it nanofiltration or ultrafiltration?

Answer: We revised the title of the article as follows:

"Al-Ani D. M., Al-Ani F. H., Alsalhy Q. F., Ibrahim S. S., Preparation and characterization of ultrafiltration membranes from PPSU-PES polymer blend for dyes removal, Chemical Engineering Comunications, published online 4 October 2019."

Reviewer 3 Report

The manuscript presents the preparation, characterization and performance evaluation of the PPSU/PES blend membranes. The experimental works and discussion are relatively well presented in detail, although the idea is not bright new. The scientific findings from this work can still contribute to the literature. Before the papers can be considered for publication, there are several comments from the reviewer for the authors to address:

  1. The trends in the membrane thickness and porosity do not match, especially for membrane with higher PES content. Judging from equation 1 and 2, the membrane thickness and porosity is inversely related while Figure 6 and 7 showed differently. Please explain.
  2. Line 324-325, the description of PWP is inappropriate since P5 actually shows higher fluxes than P3 and P4. Please revise.
  3. Following comments 2, the permeation fluxes with dye feed in Figure 9 shows a different trend to PWP results in Figure 8. The permeation flux of 0ppm dye feed is effectively PWP, but the data are different. Please clarify.
  4. Figure 12 may be used as a graphical abstract; its purpose is unclear at the end of main text.

Minor corrections:

  1. Line 154, the unit of PWP should be L/(m2hbar) calculating from equation 3 instead of kg/(m2hbar);
  2. An equal sign is missing from equation 5;
  3. Line 206, ‘Table 2 presents…’, the table number is wrong; and please make sure all other figure numbers and tables numbers are correct.
  4. Figure 2, it’s better to use P1 to P5 for the notation of the FTIR spectra;
  5. Line 293 is redundant as it repeats the previous paragraph;
  6. Figure 6, what is the physical meaning of the horizontal error bars on the porosity curve?
  7. Line 356, Figure 10 should be Figure 11.

Author Response

Dear Editor

I would like to submit a revised form manuscript entitled “Removal of dye from a leather tanning factory by flat-sheet blend ultrafiltration (UF) membrane" for consideration for publication in the Membranes. This article is revised according to the reviewer comments. I appreciate the effort of the reviewer to improve our article, thank you. Please answer to the reviewer comments presented below.

Your consideration for this manuscript with revised form is highly appreciated.

Sincerely

Prof. Dr. Qusay F. Alsalhy

Director of Membrane Technology Research Unit

Chemical Engineering Department

University of Technology,

Alsinaa Street No. 52

Baghdad, Iraq

Email:qusay_alsalhy@yahoo.com

80006@uotechnology.edu.iq

Answer to the reviewer's comments:

Reviewer 3

The manuscript presents the preparation, characterization and performance evaluation of the PPSU/PES blend membranes. The experimental works and discussion are relatively well presented in detail, although the idea is not bright new. The scientific findings from this work can still contribute to the literature. Before the papers can be considered for publication, there are several comments from the reviewer for the authors to address:

  1. The trends in the membrane thickness and porosity do not match, especially for membrane with higher PES content. Judging from equation 1 and 2, the membrane thickness and porosity is inversely related while Figure 6 and 7 showed differently. Please explain.

Answer: Thanks to the reviewer for this important comment; therefore we revised and added the following sentences:

"Increasing the PES to 3 or 4% caused a trend that slightly decreased the membrane thickness. This may be due to the hydrophilic nature of the PES in the PPSU matrix, which enhanced the solution’s thermodynamic instability in the phase inversion process and also increased porosity. When the PES concentration increased from 1 to 2%, there was a sudden jump in the porosity value, possibly because of the increase in the number of fingerlike shape………….

Also, the difference in interaction parameters of two polymers with solvent may lead to aggregation of PES in PPSU-PES at high PES loading during membrane formation and the results presented in Table 2 confirm this behavior. Thus, it can be realized that the relation between the membrane thickness and porosity were influence by the polymer properties and its sympathy and interaction parameters between the solvent and polymer. Besides, increasing the amount of the hydrophilic polymer could affect the trends in membrane thickness and porosity."

  1. Line 324-325, the description of PWP is inappropriate since P5 actually shows higher fluxes than P3 and P4. Please revise.

Answer: We agree with the reviewer comment, therefore we revised the following sentence:

On the other hand, increasing the PES concentration beyond 4% led to a reduction in the PWP, to about 67.25 (L/m2. h). Revised to " On the other hand, increasing the PES concentration beyond 4% led to increase in the PWP, to about 67.25 (L/m2. h).

  1. Following comments 2, the permeation fluxes with dye feed in Figure 9 shows a different trend to PWP results in Figure 8. The permeation flux of 0ppm dye feed is effectively PWP, but the data are different. Please clarify.

Answer: Figure 9 was not the results of the current work, for this reason we did not present its discussed in the manuscript. Therefore, we delete it from the paper. We are apologize to the reviewers.

  1. Figure 12 may be used as a graphical abstract; its purpose is unclear at the end of main text.

Answer: The Membranes Journal did not have the Graphical abstract in the template form or the contexts of the papers and the purpose of Figure 12 (revised to Figure 11) is to explain the overall procedures and results of the present work starting from the preparation of the membranes down to measuring the performance of the membranes as it is common in all Journals.

Minor corrections:

  1. Line 154, the unit of PWP should be L/(m2hbar) calculating from equation 3 instead of kg/(m2hbar);

Answer: We revised the unit of PWP from kg/(m2hbar) to L/(m2hbar).

  1. An equal sign is missing from equation 5;

Answer: We added the missing equal sign in Equation 5

  1. Line 206, ‘Table 2 presents…’, the table number is wrong; and please make sure all other figure numbers and tables numbers are correct.

Answer: We revised number and location of all Tables and Figures in the text and caption carefully, thanks for this comment

  1. Figure 2, it’s better to use P1 to P5 for the notation of the FTIR spectra;

Answer: we revised the notation of FTIR spectra from P1 to P5.

  1. Line 293 is redundant as it repeats the previous paragraph;

Answer: We delete it from the section

  1. Figure 6, what is the physical meaning of the horizontal error bars on the porosity curve?

Answer: We deleted the horizontal error bars because it is redundant 

  1. Line 356, Figure 10 should be Figure 11.

Answer: we revised all the number of Figures

Round 2

Reviewer 1 Report

Scheme 1, the dye molecule should be charge neutral.

Reviewer 2 Report

The authors have addressed all my comments thoroughly and provided appropriate explanations. I believe the manuscript has a much better flow of information with detailed discussions, proper descriptions of figures and tables, and conclusion drawn from the results obtained. I recommend the manuscript should be considered for publication. 

Reviewer 3 Report

The authors have addressed the reviewer's comments satisfactorily and revised the manuscript well. The manuscript is now recommended for publiction in Membranes.